# Interaction of Human ACE2 to Membrane-Bound SARS-CoV-1 and SARS-CoV-2 S Glycoproteins

**DOI:** 10.3390/v12101104

**Published:** 2020-09-29

**Authors:** Sai Priya Anand, Yaozong Chen, Jérémie Prévost, Romain Gasser, Guillaume Beaudoin-Bussières, Cameron F. Abrams, Marzena Pazgier, Andrés Finzi

**Affiliations:** 1Centre de Recherche du CHUM, Montréal, QC H2X 0A9, Canada; sai.anand@mail.mcgill.ca (S.P.A.); jeremie.prevost@umontreal.ca (J.P.); romain.gasser@umontreal.ca (R.G.); guillaume.beaudoin-bussieres@umontreal.ca (G.B.-B.); 2Department of Microbiology and Immunology, McGill University, Montréal, QC H3A 2B4, Canada; 3Infectious Disease Division, Department of Medicine, Uniformed Services University of the Health Sciences, Bethesda, MD 20814–4712, USA; yaozong.chen.ctr@usuhs.edu (Y.C.); marzena.pazgier@usuhs.edu (M.P.); 4Departement de Microbiologie, Infectiologie et Immunologie, Université de Montréal, Montréal, QC H2X 0A9, Canada; 5Department of Biochemistry and Molecular Biology, Drexel University College of Medicine, Philadelphia, PA 19104, USA; cfa22@drexel.edu

**Keywords:** Coronavirus, SARS-CoV-1, SARS-CoV-2, spike glycoproteins, human ACE2 receptor, ACE2-Fc, CR3022 antibody, neutralization, COVID-19

## Abstract

Severe acute respiratory syndrome virus 2 (SARS-CoV-2) is responsible for the current global coronavirus disease 2019 (COVID-19) pandemic, infecting millions of people and causing hundreds of thousands of deaths. The viral entry of SARS-CoV-2 depends on an interaction between the receptor-binding domain of its trimeric spike glycoprotein and the human angiotensin-converting enzyme 2 (ACE2) receptor. A better understanding of the spike/ACE2 interaction is still required to design anti-SARS-CoV-2 therapeutics. Here, we investigated the degree of cooperativity of ACE2 within both the SARS-CoV-2 and the closely related SARS-CoV-1 membrane-bound S glycoproteins. We show that there exist differential inter-protomer conformational transitions between both spike trimers. Interestingly, the SARS-CoV-2 spike exhibits a positive cooperativity for monomeric soluble ACE2 binding when compared to the SARS-CoV-1 spike, which might have more structural restraints. Our findings can be of importance in the development of therapeutics that block the spike/ACE2 interaction.

## 1. Introduction

Severe acute respiratory syndrome virus 2 (SARS-CoV-2) is the cause of the current and rapidly evolving coronavirus disease 2019 (COVID-19) pandemic. One potential therapeutic target receiving significant attention is the interaction between the SARS-CoV-2 spike (S) glycoprotein and its receptor, human angiotensin-converting enzyme 2 (ACE2). The S glycoprotein is a heavily glycosylated type I membrane protein present as a trimer on mature virions and consists of three S1/S2 heterodimers [1,2]. The S1 subunit contains a receptor-binding domain (RBD) that specifically binds to ACE2 and can undergo hinge-like movements to switch between an “up” position for receptor binding and “down” position for potential immune evasion [1,2,3,4]. The S glycoprotein can only bind to ACE2 with the RBD in the “up” state and this results in the dissociation of the trimer [5,6]. ACE2, the primary receptor for both the SARS-CoV-1 and SARS-CoV-2 viruses [7,8], is a type I membrane protein [9] and the soluble version of ACE2 has been shown to bind both S glycoproteins and block viral entry [10]. Currently, recombinant human ACE2 is being tested as a treatment option for patients with COVID-19 to decrease viral replication (NCT04335136). 

Phylogenetic analyses have demonstrated that SARS-CoV-2 and SARS-CoV-1 are closely related, with an ~80% genomic sequence identity [11,12]. Moreover, recent studies have also compared the spike glycoproteins of SARS-CoV-1 and SARS-CoV-2, with a 76% amino acid sequence identity between the two and a 74% amino acid sequence identity between their RBDs [13] which directly contribute to the engagement of ACE2. Available structural and functional data reveal several similarities in how both interact with ACE2: first, the contact interface of ACE2 and the RBDs of the two spikes are largely similar [6]; second, ACE2 binding epitopes on both RBDs are inaccessible in the fully closed spike conformation [6,14]; third, effective receptor engagement requires both the “up” orientation and a slight rotation of the RBD [15]; and fourth, in the ACE2-free condition, spike trimers on the virion surface can equilibrate between closed (3-RBD-down) and open (one or more RBDs up) states [1,2,15,16]. On the other hand, the existing structural data for SARS-CoV-1 and the recent information that has become available in the last few months regarding the architecture and conformational status of the soluble or virion bound SARS-CoV-2 spike point toward important differences in their thermodynamics. The affinity of SARS-CoV-1 S and SARS-CoV-2 S glycoproteins for soluble monomeric ACE2 (sACE2) has been determined, with the latter having a 10- to 20-fold higher binding affinity. It has been suggested that this could be a critical factor explaining the higher transmissibility of SARS-CoV-2 [1,6,17]. Additionally, recent studies have characterized the nature of the interaction between an engineered dimeric ACE2-Fc fusion protein and the trimeric SARS-CoV-2 S proteins, showing a high-affinity interaction superior to that seen with sACE2 [18,19]. ACE2-Fc is also able to neutralize SARS-CoV-2 S more efficiently than SARS-CoV-1 S [20]. In this study, we attempted to better understand the interaction between sACE2 or ACE2-Fc and the trimeric membrane-bound SARS-CoV-1 S or SARS-CoV-2 S glycoproteins by evaluating the cooperative binding of each of the ligands and receptors. Our results further highlight conformational differences between the SARS-CoV-1 and SARS-CoV-2 spike glycoproteins.

## 2. Material and Methods 

### 2.1. Plasmids

The plasmids expressing the different human coronavirus spikes (SARS-CoV-2 and SARS-CoV-1) were previously reported [7]. The D614G mutation was introduced using the QuikChange II XL site-directed mutagenesis protocol (Agilent Technologies, Santa Clara, CA, USA). The presence of the desired mutations was determined by automated DNA sequencing. The plasmid encoding for soluble human ACE2 (residues 1–615) fused with an 8XHisTag was reported elsewhere [2]. To generate the recombinant ACE2-Fc fusion plasmid, DNA encoding ACE2 (1–615) was linked to Fc segment of human IgG1 and the whole fusion fragment was cloned into pACP-tag(m)-2 vector using NheI/NotI as restriction sites. The vesicular stomatitis virus G (VSV-G)-encoding plasmid (pSVCMV-IN-VSV-G) was previously described [21]. 

### 2.2. Cell Lines 

293T human embryonic kidney cells (obtained from ATCC, Manassas, VA, USA) were maintained at 37 °C under 5% CO_2_ in Dulbecco’s modified Eagle’s medium (DMEM) (Wisent, St. Bruno, QC, Canada) containing 5% fetal bovine serum (VWR) and 100 μg/mL of penicillin-streptomycin (Wisent). The generation and maintenance of 293T-ACE2 cell line were previously reported [22]. 

### 2.3. Protein Expression and Purification

FreeStyle 293F cells (Invitrogen, Rockford, IL, USA) were grown in FreeStyle 293F medium (Invitrogen) to a density of 1 × 10^6^ cells/mL at 37 °C with 8% CO_2_ with regular agitation (150 rpm). Cells were transfected with a plasmid coding for soluble ACE2 or ACE2-Fc using ExpiFectamine 293 transfection reagent, as directed by the manufacturer (Invitrogen). One week later, cells were pelleted and discarded. Supernatants were filtered using a 0.22 µm filter (Thermo Fisher Scientific, Waltham, MA, USA). The recombinant sACE2 protein was purified by nickel affinity columns (Invitrogen) and ACE2-Fc was purified using protein A affinity column (Cytiva, Marlborough, MA, USA), as directed by the manufacturers. The protein preparations were dialyzed against phosphate-buffered saline (PBS) and stored in aliquots at −80 °C until further use. To assess purity, recombinant proteins were loaded on SDS-PAGE gels and stained with Coomassie blue.

### 2.4. Cell Surface Staining and Flow Cytometry Analysis

Using the standard calcium phosphate method, 10 μg of Spike expressor and 2 μg of a green fluorescent protein (GFP) expressor (pIRES-GFP) were transfected into 2 × 10^6^ 293T cells. To determine the Hill coefficients, cells were preincubated with increasing concentrations of soluble ACE2 (0 to 11,500 nM), ACE2-Fc (0 to 500 nM), or the monoclonal antibody CR3022 (0 to 270 nM) 48 h post-transfection. sACE2 binding was detected using a polyclonal goat anti-ACE2 (RND systems). AlexaFluor-647-conjugated goat anti-human IgG (H+L) Ab (Invitrogen) and AlexaFluor-647-conjugated donkey anti-goat IgG (H+L) Ab (Invitrogen) were used as secondary antibodies. The percentage of transfected cells (GFP+ cells) was determined by gating the living cell population based on viability dye staining (Aqua Vivid, Invitrogen). Samples were acquired on an LSRII cytometer (BD Biosciences, Mississauga, ON, Canada) and data analysis was performed using FlowJo vX.0.7 (Tree Star, Ashland, OR, USA). Hill coefficient analyses were done using GraphPad Prism version 8.0.1 (GraphPad, San Diego, CA, USA). 

### 2.5. Virus Neutralization Assay 

Target cells were infected with single-round luciferase-expressing lentiviral particles. Briefly, 293T cells were transfected by the calcium phosphate method with the lentiviral vector pNL4.3 R-E- Luc (NIH AIDS Reagent Program) and a plasmid encoding for SARS-CoV-2 spike (WT or D614G), SARS-CoV-1 spike, or VSV-G at a ratio of 5:4. Two days after transfection, the cell supernatants were harvested. Each virus preparation was frozen and stored in aliquots at −80 °C until use. 293T-ACE2 target cells were seeded at a density of 1 × 10^4^ cells/well in 96-well luminometer-compatible tissue culture plates (Perkin Elmer, Waltham, MA, USA) 24 h before infection. Luciferase-expressing recombinant viruses in a final volume of 100 μL were incubated with increasing concentrations of soluble ACE2 (0 to 11,500 nM), ACE2-Fc (0 to 500 nM), or the monoclonal antibody CR3022 (0 to 270 nM) for 1 h at 37 °C and were then added to the target cells for an additional 4 hours followed by incubation for 48 h at 37 °C; the medium was then removed from each well, and the cells were lysed by the addition of 30 μL of passive lysis buffer (Promega, Madison, WI, USA) followed by three freeze–thaw cycles. An LB 941 TriStar luminometer (Berthold Technologies, Bad Wildbad, Germany) was used to measure the luciferase activity of each well after the addition of 100 μL of luciferin buffer (15 mM MgSO4, 15 mM KPO4 [pH 7.8], 1 mM ATP, and 1 mM dithiothreitol) and 50 μL of 1 mM d-luciferin potassium salt (Prolume, Pinetop, AZ, USA). The neutralization half-maximal inhibitory concentration (IC_50_) represents the ligand concentration required to inhibit 50% of the infection of 293T-ACE2 cells by recombinant lentiviral viruses bearing the indicated surface glycoproteins. IC_50_ values were determined using a normalized non-linear regression using Graphpad Prism software.

## 3. Results and Discussion 

### 3.1. Differences Between SARS-CoV-1 S and SARS-CoV-2 Spikes in Their Abilities to Engage sACE2, ACE2-Fc, and CR3022

To better understand the interactions between membrane-bound SARS-CoV-1 and SARS-CoV-2 S glycoproteins with their receptor, human ACE2, we sought to determine the cooperativity of ACE2 within the respective trimers. To assess this, we calculated the Hill coefficient, which is the steepness of a concentration–response curve and reflects the degree of cooperativity between a ligand and its receptor [23,24]. Briefly, HEK293T cells were transfected with plasmids expressing the full-length SARS-CoV-1 S and SARS-CoV-2 S glycoproteins. We also tested the SARS-CoV-2 S D614G mutant that is associated with higher infectivity and is now the strain circulating worldwide [25,26,27]. Transfected cells were incubated with increasing concentrations of sACE2 and bound sACE2 was revealed with an anti-ACE2 antibody. These results were used to calculate the Hill coefficient as indicated in Material and Methods. Both the SARS-CoV-2 S and its D614G counterpart demonstrated a positive cooperativity of sACE2 binding (Hill coefficient > 1) (Figure 1A), thus, indicating that the D614G mutation does not overly affect S conformation, at least regarding sACE2 interaction, in line with recent findings [26,27]. Interestingly, SARS-CoV-1 S presented negative cooperativity (Hill coefficient < 1), suggesting that the interaction of sACE2 with one SARS-CoV-1 S protomer reduces the efficiency with which additional sACE2 molecules can engage adjacent S protomers. To evaluate if the differential Hill coefficients observed between SARS-CoV-2 and SARS-CoV-1 was conserved among different RBD ligands, we tested two additional RBD-binding ligands: ACE2-Fc, a molecule presenting two ACE2 domains (residues 1–615) fused to a Fc fragment, and the CR3022 monoclonal antibody, which is specific to the SARS-CoV-1 RBD and has been shown to cross-react strongly with the RBD of SARS-CoV-2 and does not compete with the binding of ACE2 [28,29]. Interestingly, no differential cooperativity between SARS-CoV-2 and SARS-CoV-1 was observed for ACE2-Fc, suggesting that the enhanced avidity provided by ACE2-Fc, which allows for multiple spike protomers to bind, is able to overcome potential structural restraints present in the SARS-CoV-1 S (Figure 1B). Of note, we observed negative cooperativity of CR3022 for all tested S glycoproteins (Figure 1C), in line with previous findings showing that the binding of CR3022 leads to the destruction of the prefusion SARS-CoV-2 S trimer [30]. Thus, it is possible that the binding of CR3022 to one RBD protomer distorts the trimer structure, preventing additional CR3022 molecules from binding.

### 3.2. Sensitivity of Viruses Harboring SARS-CoV-1 S and SARS-CoV-2 Spikes to Neutralization by sACE2, ACE2-Fc, and CR3022

Next, we tested the abilities of sACE2 and ACE2-Fc to neutralize SARS-CoV-1 and SARS-CoV-2 spike-bearing pseudovirions. We observed a higher neutralization potency of sACE2 against SARS-CoV-2 S (wt or D614G) when compared to SARS-CoV-1 (Table 1). However, whether the negative cooperativity observed for the SARS-CoV-1 S/sACE2 interaction could explain the ~5-fold lower neutralization potency of monomeric ACE2 when compared to SARS-CoV-2 (IC_50_ = 245 nM for SARS-CoV-2 S; 1359 nM for SARS-CoV-1 S) (Figure 2A; Table 1) remains to be determined. In agreement with previous reports [18], ACE2-Fc neutralized both SARS-CoV-1 S and SARS-CoV-2 S with a higher efficiency compared to monomeric sACE2 (Figure 2B; Table 1), further supporting previous observations that ligand multimerization enhances potency by providing higher avidity [18,20,31,32]. Interestingly, we observed that sACE2 IC_50_ was only reached upon its half-maximal binding to the respective S proteins, suggesting a model where the occupancy of at least two or more protomers of the SARS-CoV-1 S or SARS-CoV-2 S by sACE2 is needed for virus neutralization (Figure 2D,E). 

### 3.3. Proposed Energy Landscapes of Spike Trimer Opening of SARS-CoV-2 and SARS-CoV-1

Cryo-EM data collected on free (receptor-unbound) SARS-CoV-2 S indicate that it assumes three distinct states. Preferential are the 3-RBD-down state (41%) and the 1-RBD-up state (45%) that exists at a near 1:1 ratio, whereas less preferential is the 2-RBD-up conformation (approximately 10%) [1,16,33]. This indicates that there is a relatively low barrier for the open–closed transition in its energy landscape. Consistent with this more easy-to-open propensity of the SARS-CoV-2 spike, Zhou et al. have identified 15% one-ACE2-bound, 43% two-ACE2-bound, and 38% three ACE2-bound SARS-CoV-2 spike by single-particle cryo-EM when mixing spike and ACE2 in a 1:3 molar ratio [15]. The higher percentage of the two/three-ACE2 bound state over the single ACE2-bound state corroborates the positive cooperativity seen between monomeric receptor and SARS-CoV-2 trimer. Interestingly, in all ACE2-bound spike trimers, RBDs not bound to ACE2 are in the “down” conformation, suggesting that the down-to-up rearrangement of the spike trimer is the rate-limiting step in receptor binding.

In contrast, available data reveal differences in the conformational dynamics of the SARS-CoV-1 spike which seems to have less propensity to engage multiple ACE2 monomers. With a ratio of 1:3 (spike:ACE2), as described by Zhou et al. for the SARS-CoV-2 spike, the prevailing complex fraction observed for SARS-CoV-1 was the one ACE2 monomer bound spike [34]. Interestingly, the cryo-EM structure of this complex revealed three distinct conformations of spike in which the loaded RBD adopted three different tilted orientations and the other two RBDs remained in the down state [34]. This is different from the multiple ACE2 loading of the SARS-CoV-2 spike which displays one essentially identical RBD up conformation. 

Based on the structural dynamic data described above we propose a model as shown in Figure 3. According to our model, the SARS-CoV-1 spike has more structural restraints from the NTD, SD1, SD2 and S2 domains and, therefore, needs to overcome a higher energy barrier to open. As a result, the SARS-CoV-1 spike has a smaller population in the open conformation in the absence of ACE2, which is in line with the unavailability of structures of multi-ACE2 bound SARS-CoV-1 spikes and our finding that monomeric sACE2 binding suppresses the opening of the other two RBDs. Interestingly, these differences can be detected only if monomeric ACE2 is used to form the complex but not with ACE2-Fc. There is still a question of debate if ACE2 monomer or dimer is involved in the entry process of both SARS-CoV-1 and SARS-CoV-2. If the latter, the differences we observe for monomeric ACE2 resulting from the intrinsic structural features of the two coronavirus spikes which leads to different energy landscapes could be mitigated during the process of viral entry in vivo. However, whether these differences are important for antibody-mediated neutralization remains to be determined. 

Altogether, our results show differential inter-protomer conformational transitions between SARS-CoV-2 and SARS-CoV-1 S glycoproteins upon sACE2 binding. A better understanding of conformational differences between the S glycoproteins between these two beta-coronaviruses might prove useful for the development of new therapeutics and/or vaccine design.

## Figures and Tables

**Figure 1 viruses-12-01104-f001:**
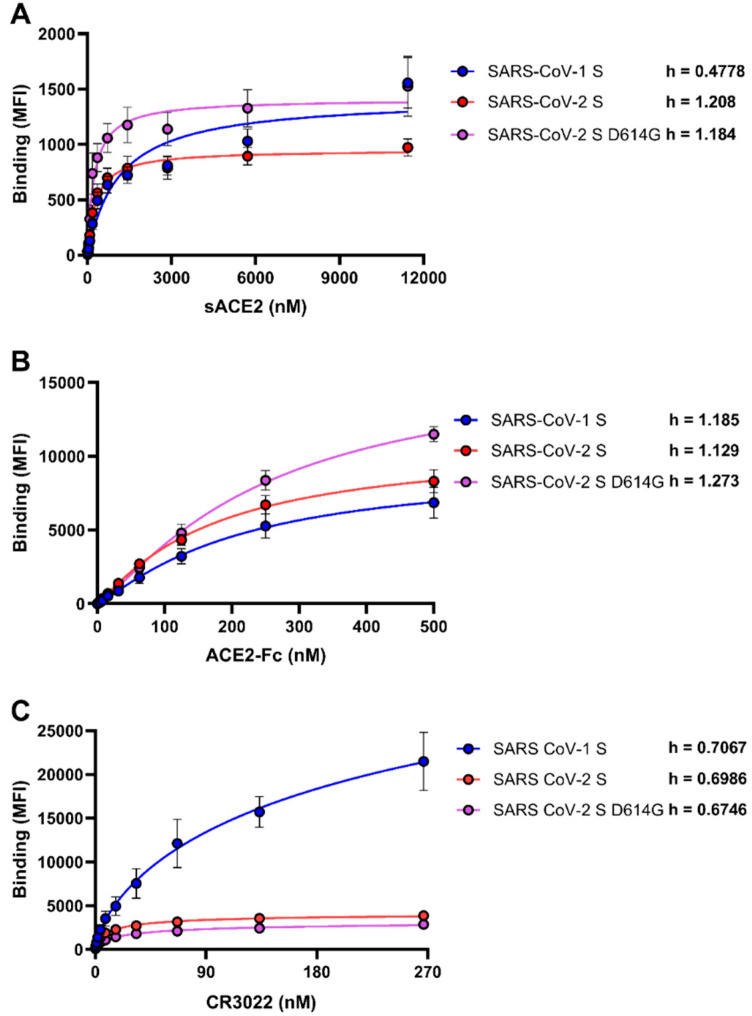
Differences between severe acute respiratory syndrome virus 1 (SARS-CoV-1) and severe acute respiratory syndrome virus 2 (SARS-CoV-2) spikes in their abilities to engage soluble monomeric angiotensin-converting enzyme 2 (sACE2), ACE2-Fc, and CR3022. The binding of (**A**) sACE2, (**B**) ACE2-Fc, and (**C**) CR3022 to SARS-CoV-1 or SARS-CoV-2 (wt or D614G) spikes expressed on the cell surface were measured by flow cytometry. Increasing concentrations of each ligand were incubated with Spike-expressing cells as described in the Material and Methods. Means ± SEM derived from at least three independent experiments are shown. The Hill coefficient was determined using GraphPad software.

**Figure 2 viruses-12-01104-f002:**
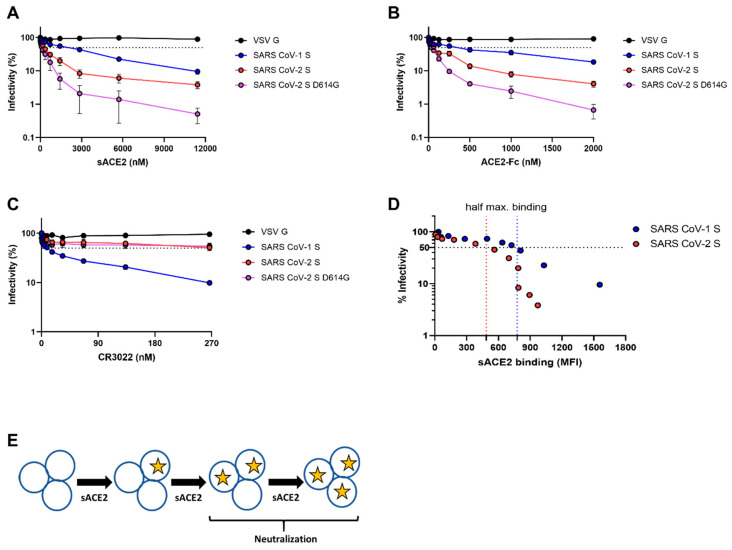
Sensitivity of viruses harboring SARS-CoV-1 S and SARS-CoV-2 spikes to neutralization by sACE2, ACE2-Fc, and CR3022. Pseudoviral particles coding for the luciferase reporter gene and bearing the following glycoproteins: SARS-CoV-2 (wt or D614G) S, SARS-CoV-1 S, or VSV-G were used to infect 293T-ACE2 cells. Pseudoviruses were incubated with increasing concentrations of (**A**) sACE2, (**B**) ACE2-Fc, and (**C**) CR3022 at 37 °C for 1 h prior to infection of 293T-ACE2 cells. Means ± SEM derived from at least two independent experiments are shown. (**D**) Neutralization by sACE2 was correlated with the sACE2 binding quantified by flow cytometry to SARS-CoV-1 and SARS-CoV-2 spikes. (**E**) Model indicating the stoichiometry needed for neutralization by sACE2 to either SARS-CoV-1 S-or SARS-CoV-2 (wt or D614G) S-bearing pseudovirions.

**Figure 3 viruses-12-01104-f003:**
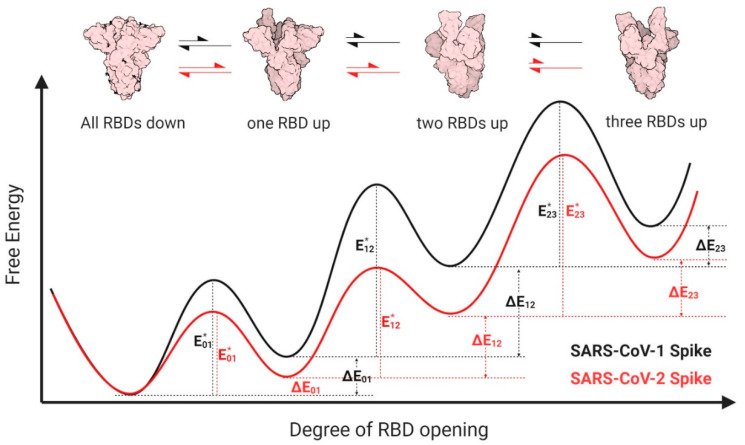
Proposed energy landscapes of spike trimer opening of SARS-CoV-2 and SARS-CoV-1. We assume conformational landscapes for both species of spikes (SARS-CoV-1: black; SARS-CoV-2: red) that permit equilibration among four distinct states: all- receptor-binding-domains (RBDs) -down (“0”), one-RBD-up (“1”), two-RBDs-up (“2”), and three-RBDs-up (“3”). We hypothesize that the SARS-CoV-2 spike energetic barrier for transitioning from all-RBDs-down to one-RBD-up ($E_{01}^{*}$) is lower for SARS-CoV-2 than for SARS-CoV-1, while both experience the same barrier for the reverse transition, making the one-RBD-up state more stable for SARS-CoV-2 than for SARS-CoV-1, as illustrated by the relative energy differences $\Delta E_{01}$. However, SARS-CoV-1 spike likely needs to overcome substantially higher energy barriers to transit from the one-RBD-up state to the two-RBDs-up state ($E_{12}^{*}$), as compared with SARS-CoV-2 spike. This underlying conformational selection mechanism results in a larger population of SARS-CoV-2 spike in two or three RBDs up state and has higher chance to engage multiple ACE2, which eventually spurs the complete open and dissociation of S1 from S2 and induces membrane fusion. Cartoon representation of closed, one-RBD-up, two-RBDs-up states of spike were generated from deposited structures in Protein Data Bank (6ZWV, 6VSB, 6 × 2B, respectively). The three-RBDs-up model was generated by C3 symmetry superposition of three one-RBD-up protomer in 6VSB.

**Table 1 viruses-12-01104-t001:** Neutralization half-maximal inhibitory concentrations (IC_50_). VSV-G—vesicular stomatitis virus G.

Pseudotype	Neutralization (IC50; nM)			
	sACE2	ACE2-Fc	CR3022	Fold Change
				(sACE2/ACE2-Fc)
**VSV G**	**>11500**	**>2000**	**>270**	**-**
**SARS CoV-1 S**	**1359**	**218.5**	**7.481**	**6.22**
**SARS CoV-2 S**	**245.4**	**56.82**	**131.1**	**4.32**
**SARS CoV-2 S D614G**	**103.8**	**28.73**	**80.25**	**3.61**
	**Legend**			
	**sACE2**	**ACE2-Fc**	**CR3022**	
	**IC50 < 1000**	**IC50 < 100**	**IC50 < 10**	
	**1000 < IC50 < 11500**	**100 < IC50 < 2000**	**10 < IC50 < 270**	
	**IC50 > 11500**	**IC50 > 2000**	**IC50 > 270**	

Red = No neutralization; Yellow = Low neutralization; Green = High neutralization.

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
