# Peer review of "Interaction of Human ACE2 to Membrane-Bound SARS-CoV-1 and SARS-CoV-2 S Glycoproteins"

_viruses, 2020, doi:10.3390/v12101104_

Round 1

Reviewer 1 Report

General comment

The study reported Interaction of human ACE2 to membrane-bound SARS-CoV-1 and SARS-CoV-2 S glycoproteins.

In general, the paper was well written and understandable. I recommend this manuscript for publication.

Specific comments

1. M&M were suitable for the research work

2. It would still be worthwhile for them to write about whether there may be age or gender differences

3. Do the results suggest any effects or differences for ACE inhibitor consumers?

4. it is also worth mentioning a few words about the following article:

Kui K. Chan, Danielle Dorosky, Preeti Sharma, Shawn A. Abbasi, John M. Dye, David M. Kranz, Andrew S. Herbert, Erik Procko. Engineering human ACE2 to optimize binding to the spike protein of SARS coronavirus 2. Science. 04 Sep 2020: Vol. 369, Issue 6508, pp. 1261-1265 DOI: 10.1126/science.abc0870

Author Response

Reviewer #1

General comment

The study reported Interaction of human ACE2 to membrane-bound SARS-CoV-1 and SARS-CoV-2 S glycoproteins.

In general, the paper was well written and understandable. I recommend this manuscript for publication.

Response: we thank the reviewer for his/her appreciation of our work

Specific comments

  1. M&M were suitable for the research work

Response: we thank the reviewer for his/her appreciation of our work

  1. It would still be worthwhile for them to write about whether there may be age or gender differences

Response: we agree with the reviewer that age and sex differences do have an impact on SARS-CoV-2 infection.  However, our manuscript is about the differences between SARS-CoV-1 and SARS-CoV-2 engagement at the molecular level, independently of sex or age differences.  No human samples were used here and therefore these variables do not apply to our work.

  1. Do the results suggest any effects or differences for ACE inhibitor consumers?

Response: our results suggest that sACE2 oligomerization might improve the potency of ACE2-based inhibitors.  This was discussed in the original version of our manuscript at lines 204-207.

  1. it is also worth mentioning a few words about the following article:

Kui K. Chan, Danielle Dorosky, Preeti Sharma, Shawn A. Abbasi, John M. Dye, David M. Kranz, Andrew S. Herbert, Erik Procko. Engineering human ACE2 to optimize binding to the spike protein of SARS coronavirus 2. Science. 04 Sep 2020: Vol. 369, Issue 6508, pp. 1261-1265 DOI: 10.1126/science.abc0870

Response: We thank the reviewer for his/her suggestion.  We agree with the above-referenced work that shows that stable dimeric variants of ACE2 are more potent at inhibiting both SARS-CoV-1 and SARS-CoV-2.  The results presented in our manuscript provides the reason of why this is necessary. This was already discussed in the original manuscript at lines 204-207 and we have now added the reference suggested by the reviewer (new reference #32).

Reviewer 2 Report

In this interesting work, the Authors pay attention to the conformational differences of the glycoproteins of the SARS CoV1 and 2 viruses and the ACE2 binding. These differences are important for therapeutic purposes in anticipation of new therapies and vaccines.
The experimental set was well designed and the results obtained very interesting. I ask the authors for a brief comment in relation to a non-negligible aspect reported by several parties (doi: 10.1186 / s13293-020-00330-7; doi: 10.1111 / bph.1520): the different response to the virus with respect to sex. Could their results partially explain this phenomenon? Reporting this comment also within the work itself.

Author Response

In this interesting work, the Authors pay attention to the conformational differences of the glycoproteins of the SARS CoV1 and 2 viruses and the ACE2 binding. These differences are important for therapeutic purposes in anticipation of new therapies and vaccines.
The experimental set was well designed and the results obtained very interesting. I ask the authors for a brief comment in relation to a non-negligible aspect reported by several parties (doi: 10.1186 / s13293-020-00330-7; doi: 10.1111 / bph.1520): the different response to the virus with respect to sex. Could their results partially explain this phenomenon? Reporting this comment also within the work itself.

Response: we thank the reviewer for his/her appreciation of our work

 We agree with the reviewer that age and sex differences do have an impact on SARS-CoV-2 infection.  However, our manuscript is about the differences between SARS-CoV-1 and SARS-CoV-2 engagement at the molecular level, independently of sex or age differences.  No human samples were used here and therefore these variables do not apply to our work.